# Phenotypic and Genotypic Screening of Colistin Resistance Associated with Emerging Pathogenic *Escherichia coli* Isolated from Poultry

**DOI:** 10.3390/vetsci9060282

**Published:** 2022-06-09

**Authors:** Heba Badr, Abdelhafez Samir, Essam Ismail El-Tokhi, Momtaz A. Shahein, Flourage M. Rady, Ashraf S. Hakim, Ehab Ali Fouad, Engy Farahat El-Sady, Samah F. Ali

**Affiliations:** 1Bacteriology Unit, Reference Laboratory for Veterinary Quality Control on Poultry Production, Animal Health Research Institute, Ministry of Agriculture, Agricultural Research Center (ARC), Nadi El-Seid Street, Dokki, P.O. Box 246, Giza 12618, Egypt; 2Biotechnology Unit, Reference Laboratory for Veterinary Quality Control on Poultry Production, Animal Health Research Institute, Ministry of Agriculture, Agricultural Research Center (ARC), Nadi El-Seid Street, Dokki, P.O. Box 246, Giza 12618, Egypt; abdelhafez_samir@yahoo.com; 3Biotechnology Department, Animal Health Research Institute, Agricultural Research Center (ARC), Giza 12618, Egypt; eltokhy_essam@yahoo.com; 4Department of Virology Research, Animal Health Research Institute, Agricultural Research Center (ARC), Giza 12618, Egypt; momtaz.shahein@yahoo.com; 5Mycology Department, Animal Health Research Institute, Agricultural Research Center (ARC), Shebin Elkom 32511, Egypt; flourageragy@gmail.com; 6Microbiology and Immunology Department, National Research Centre, 33 Bohouth st., Dokki, Cairo 12622, Egypt; migris410@yahoo.com (A.S.H.); ehabfoaud@gmail.com (E.A.F.); elsadyengy@gmail.com (E.F.E.-S.); 7Bacteriology Department, Animal Health Research Institute (AHRI), Agricultural Research Center (ARC), Giza 12618, Egypt; samah_hefny2004@yahoo.com

**Keywords:** APEC, colistin resistance, *mcr*-1, Egypt

## Abstract

Chickens continue to be an important reservoir of zoonotic multidrug-resistant illnesses. Antimicrobial resistance correlated with colistin has emerged as a critical concern worldwide in the veterinary field and the public health sector. The current study investigated the prevalence of multidrug-resistant avian pathogenic *Escherichia coli* among chicken farms in three Egyptian governorates, focusing on colistin resistance assessment. A total of 56 *Escherichia coli* isolates were recovered out of 120 pooled samples obtained from diseased chicken broilers (46.7%). The *E. coli* isolates were serotyped to nine different serotypes; the highest incidence was for O125 (*n* = 18). The *E. coli* isolates demonstrated multidrug-resistant patterns against 10 antibiotics, especially clindamycin, tetracycline, streptomycin and ampicillin, by 100, 100, 96.4 and 92.9%, respectively. On the other hand, colistin resistance was 41.1% using AST. All *E. coli* isolates displayed positive colistin resistance growth on chromogenic medium, but only 25% represented this positivity via MIC estimation and Sensititre kit. PCR results revealed that all isolates harbored *mcr*-1, but no isolates harbored the other 2–5 *mcr* genes. In conclusion, the study demonstrated the emergence of multidrug-resistant, especially colistin-resistant, *E. coli* among chicken broiler flocks, and *mcr*-1 is the master gene of the colistin resistance feature.

## 1. Introduction

Currently, avian pathogenic *E. coli* (APEC) is a significant issue due to its pathogenicity and implication in poultry deaths. On the other side, it threatens public health via the harboring and transferring of antibiotic resistance genes around the world. The production birds (broilers, layers, breeding flocks, ducks and geese) infected with APEC usually show colibacillosis and colisepticemia signs, such as airsacculitis, cellulitis, omphailitis, pericarditis, perihepatitis, swollen-head syndrome, and other colibacillosis manifestations [1]. The antimicrobial resistance phenomenon (AMR) among avian *E. coli* isolates has become a universal health apprehension in the last decades. Antimicrobial usage in poultry production has been well reported as a likely source of AMR in humans via either a direct transfer of AMR bacteria or the horizontal transfer of the genes responsible for antimicrobial resistance (ARGs) to human pathogens [2]. Colistin is one of the most often used antimicrobials in veterinary practice, particularly in underdeveloped nations such as Egypt, where excessive antimicrobial usage (AMU) is typically unregulated [3].

Colistin (polymyxin E) shares a high degree of structural resemblance with well-known polymyxin B. It consists of a cationic peptide ring of seven amino acids linked to a hydrophobic acyl tail by a linear series of three amino acids, which plays a role in its mode of action [4]. Colistin is promptly bactericidal on Gram-negative bacteria in vitro but is considerably less effective in vivo. This shortage may be overcome by combination with certain other antibiotics in therapeutic programs [5].

However, it has been clearly affirmed that colistin’s combination with the lipopolysaccharide (LPS) layer on Gram-negative bacteria’s surface causes the outer membrane (OM) to be disrupted, and consequently, cell lysis and bacterial death occur. It is suggested that the positively charged peptide ring and hydrophobic tail structure produce a detergent-like activity and lead to damage of the cell membrane’s phospholipid bilayer [6]. In human medicine, the development of resistance to colistin is worrisome because the drug is considered a last resort for the treatment of grave illnesses caused by multi-drug-resistant *Enterobacteriaceae* bacteria [7].

Prior to 2015, chromosomal mutations were thought to be the only cause of the induction of colistin resistance. Later, many reports from various countries recorded resistance in other ways, and plasmid-mediated colistin resistance and nine *mcr* genes (*mcr*-1 to *mcr*-9) with some variations have since been found [8].

International attention has been directed to the one-health concept and the potential of livestock zoonotic, drug-resistant bacteria as a human health hazard. This study aimed to investigate the existence of *mcr*-1 as well as other *mcr* genes associated with the emerging colistin resistance among *E. coli* isolates obtained from infected broilers from some farms distributed in the middle and delta regions in Egypt and the relation between this form of genetic detection and other phenotypic methods.

## 2. Material and Methods

### 2.1. Sample Collection

The gathered samples were taken from 120 poultry farms located in various geographical regions in Egypt’s middle and delta regions (Dakahlia, Sharqia and Giza Governorates) from October 2019 to October 2020. Five diseased broiler chickens aged from 1 to 5 weeks were gathered from each farm and pooled to constitute one sample. The sampled chickens manifested depression, loss of appetite and ruffled feathers.

Birds were transmitted to the Reference Laboratory for Veterinary Quality Control on Poultry Production, AHRI, Giza, Egypt. The taken samples were exposed to post mortem inspection under aseptic conditions. Samples were taken from internal organs (heart, liver, lung and spleen) of birds that displayed colisepticemia, air sacculitis, pericarditis and perihepatitis, then pooled together for bacterial examination and isolation. All sample collection steps were legally consented by the Ethics Committee of the Animal Health Research Institute, Egypt, under agreement number (AHRI-42429).

### 2.2. Isolation and Identification

*E. coli* was recovered and identified as depicted [1]. Concisely, samples were inoculated into buffer peptone water and incubated aerobically for 24 h at 37 °C. Each incubated sample was streaked onto MacConkey agar (Oxoid, Manchester, UK) and eosin methylene blue agar (Lioflichem, Roseto degli Abruzzi, Italy) plates and incubated for 24 h at 37 °C. On MacConkey agar, the suspected colonies were 1–2 mm in diameter and had a hot-pink color, whereas on eosin methylene blue agar, they had a metallic color and sheen.

Suspected pure *E. coli* colonies were collected and subjected to further classical biochemical screening (oxidase test, citrate utilization, indole test, methyl red, Voges Proskauer and Triple Sugar Iron “TSI”) as well as using the API 20E system (BioMérieux, Craponne, France) following the manufacturer’s guidelines. Furthermore, antisera against somatic (O) antigens (DENKA SEIKEN Co., Tokyo, Japan) were used for serotyping isolated *E. coli* following the kit instructions.

### 2.3. Antimicrobial and Colistin Susceptability Patterns of E. coli Isolates

#### 2.3.1. Antimicrobial Sensitivity Test (AST)

All *E.coli* isolates were subjected to AST using the disc diffusion test on Mueller–Hinton agar, as previously described [9], against 10 antibiotics (Oxoid, Basingstoke, UK), and inhibition zones were interpreted following the guidelines of the Clinical and Laboratory Standards Institute (CLSI) [10]. The following antibiotic discs were used: ampicillin (AMP) 10 μg, apramycin (APR) 30μg, ciprofloxacin (CIP) 5 μg, clindamycin (DA) 2 μg, colistin sulphate (CT) 10 μg, norfloxacin (NOR) 10 μg, spectinomycin (SPT) 100 μg, streptomycin (S) 10 μg, sulfamethoxazole–trimethoprim (SXT) 23.75 + 1.25 = 25 μg and tetracycline (TE) 30 μg.

#### 2.3.2. Assessment of Colistin Resistance by Chromogenic Medium

The obtained isolates were streaked onto CHROMagar^TM^ COL-*APSE* base (CHROMagar, Paris, France). The chromogenic medium was used to detect and isolate colistin-resistant Gram-negative pathogens. The positive colistin resistance *E. coli* growth is represented by dark pink to reddish colonies.

#### 2.3.3. Quantitative Determination of Colistin Resistance (E-Test)

All *E. coli* isolates were checked to quantitatively determine colistin minimum inhibitory concentration (MIC). Colistin Ezy MIC strip (CL) (0.016–256 µg/mL) (Himedia^®^, Mumbai, India) was applied according to the kit instuctions. The MIC reading was interpreted following CLSI guidelines [10], with ≤2 µg/mL indicating sensitivity and ≥4 µg/mL indicating resistance. 

#### 2.3.4. Quantitative Determination of Colistin by Minimum Inhibitory Concentration (MIC)

Further testing was carried out for confirmation of MIC results on randomly selected *E. coli* isolates (n ≈ 20) shoing different degrees of colistin resistance in E-test using Sensititre^TM^ GNX3F kit (Thermo scientific, Loughborough, UK), which is a commercial broth microdilution kit of a 96-well plate coated with different concentrations of colistin (0.25, 0.5, 1, 2 and 4 µg/mL). The concentration reading depends on the sediment appearing on the bottom of the 96-well plate; when no sedement appears in all concentrations, it means MIC ≤0.25 µg/mL and the result was measured for the last sediment appearing. Of note, isolates with an MIC value of ≤2 µg/mL were deemed sensitive while those with ≥4 µg/mL were counted as resistant, following CLSI guidelines [10].

#### 2.3.5. Genotypic Characterizations of Colistin Resistance of *E. coli* Isolates

Genomic DNA was reduced from cultured bacteria via Purification Kit (Roche Diagnostics, Mannheim, Germany) following the manufacturer’s instructions. Consicely, at 56 °C/10 min, a 200 µL sample bacterial culture was treated with 10 µL of proteinase K and 200 µL of lysis buffer. After incubation, the lysate was given 200 µL of 100% ethanol. After washing and centrifuging the sample, the nucleic acid was eluted with 100 µL of elution buffer. All *E. coli* isolates were tested using primers specific for the existence of colistin resistance genes *mcr-*1, *mcr-*2, *mcr-*3, *mcr-*4 and *mcr-*5, supplied from Metabion (Planegg, Germany), as previously described and listed in Table 1. The previously extracted DNA underwent conventional PCR assays using EmeraldAmp Max PCR Master Mix (Takara, Japan). Amplification was carried out using an Applied biosystem 2720 thermal cycler. Each cycle consisted of a primary denaturation (1 cycle) for 5 min at 94 °C, except for *mcr-3* at 95 °C. All subsequent steps are illustrated in Table 1. The PCR products were electrophoresed in 1× TBE buffer at room temperature using 5 V/cm gradients on 1.5% agarose gel (Applichem GmbH, Darmstadt, Germany). For gel analysis, 20 µL of the products were placed in each gel slot. The fragment sizes were defined using a GelPilot 100 bp plus DNA Ladder (Qiagen GmbH, Hilden, Germany) and 100 bp DNA ladder (Genedirex, Taoyuan, Taiwan). A gel documentation system (Alpha Innotech, Biometra, Jena, Germany) was used to photograph the gel, and the data was processed via computer software.

## 3. Results

### 3.1. E. coli Isolation, Biochemical Characterization and Serotyping

In total, 56 out of 120 samples indicated that their farms were positive for *E.*
*coli*, with a recovery percentage of 46.7%. The *E. coli* isolates were accounted for from the pooled internal organs (heart, liver, lung and spleen) of ill broiler chickens gathered from poultry farms located in the Giza, Dakahlia and Sharqia governorates from October 2019 to October 2020. On MacConkey agar, the suspected colonies were 1–2 mm in diameter and of a hot-pink color, whereas on eosin methylene blue agar, they had a metallic color and sheen. One pure colony from each positive sample was further biochemically identified. The isolates demonstrated acidic slant with gas production bottom on TSI, positive reaction for catalase, indole and methyl red while negative for oxidase, VP and citrate production tests, with the biochemical profiles obtained via the API 20E system, as displayed in Figure 1.

The *E. coli* isolates were serotyped to nine different serotypes; the isolate with the highest incidence was O125 (*n* = 18), followed by O55 (*n* = 8) and O86 (*n* = 7), while the other serotypes were O111 (*n* = 6), O127 (*n* = 5), O159 (*n* = 4), O18 (*n* = 4) O157 (*n* = 2) and O166 (*n* = 2).

### 3.2. Antimicrobial Senstivity Patterns of the Isolated E. coli

#### 3.2.1. Antimicrobial Sensitivity Test (AST)

In our study, the various isolates were tested for their susceptibility to groups of common, commercially used antibiotics. The *E. coli* isolates showed a high resistance pattern against clindamycin (56/56, 100%) and tetracycline (56/56, 100%), followed by streptomycin (54/56, 96.4%) and ampicillin (52/56, 92.9%). At the same time, colistin resistance was (23/56, 41.1%). The complete susceptibility pattern is shown in Table 2.

#### 3.2.2. Growth on Chromogenic Medium

All 56 tested *E. coli* isolates appeared to have dark pink to reddish colonies on CHROMagar^TM^ COL-*APSE*, so it can be concluded that all isolates were colistin resistant.

#### 3.2.3. Quantitative Determination of Colistin Resistance MIC by E-Test

Upon examination of 56 *E. coli* isolates using Colistin Ezy MIC strip (CL) (0.016–256 µg/mL), only 14 isolates displayed resistance to colistin (MIC ≥ 4 µg/mL) with a percentage of 25% or greater ( Figure 2).

#### 3.2.4. Confirmatory Quantitative Determination of Colistin Resistance MIC by Sensititre

The data of the Sensititre kit for the examined 20 *E. coli* isolates showed great similarity with that obtained by E- test. The same results were obtained in 19 out of 20 isolates (95%), as shown in Table 3.

#### 3.2.5. Genotypic Characterizations of Colistin Resistance

All 56 *E. coli* isolates were positive only for the *mcr-*1 gene and negative for the rest of the colistin-resistance genes, as shown in Figure 3.

## 4. Discussion

This study investigated the incidence of pathogenic multidrug resistant *E. coli* among Egyptian farms in the Dakahlia, Giza and Sharqia governorates, highlighting the colistin resistance profiles. Overall, 56 out of 120 pooled samples showed positive isolation and biochemical identification of *E. coli*, with an incidence of 46.7%. Our result may be higher than the usual average (9.52 to 36.73%) mentioned by Lutful Kabir [16], who stated that the most susceptible age was 4 to 6 weeks. For instance, in the United States, it is reported that APEC was recovered from at least 30% of commercial flocks at every point in time [17]. Notably, the higher-than-average incidences may be common in developing countries such as Egypt. Awad et al. [18] reported a near prevalence of 51.85%. Additionally, Mohamed et al. [19] isolated 92 isolates from the internal organs of dead broiler chickens and also recovered 32 isolates from the cloacal swabs or feces of apparently healthy broiler chickens. The samples were gathered from six Algerian veterinary clinics. Another very high incidence of APEC isolates (89.2%) was reported in Pakistan [20]. Our data revealed that the *E. coli* isolates were typed serologically to nine different serotypes. with the serotype with the highest incidence being O125 (*n* = 18), followed by O55 (*n* = 8) and O86 (*n* = 7), while the other serotypes were O111 (*n* = 6), O127 (*n* = 5), O159 (*n* = 4), O18 (*n* = 4), O157 (*n* = 2) and O166 (*n* = 2). These findings coincide with previous studies [18,21] but differ from those isolated in Jordan (O1, O2, O25 and O78) [22]. Multiple *E. coli* serotypes, specified as “avian pathogenic *E. coli*” (APEC), are the etiology of colibacillosis, which is considered one of the major causes implicated in mortality (up to 20%) as well as morbidity and which results in the lowering of meat and egg production among poultry flocks worldwide [23]. Certain APEC serotypes have been reported as a prospective foodborne zoonotic pathogens as well as a reservoir or source of extra-intestinal infections in humans (i.e., the O18, O55, O125 and, of special concern, O157 serogroups) [24].

The investigated *E. coli* isolates displayed multidrug resistance behavior against the tested antibiotics with different patterns, as shown in Table 2. The high resistance to clindamycin, tetracycline, streptomycin, ampicillin and sulfamethoxazole–trimethoprim was mentioned in other reports [18,20,22,25,26,27].

Concerning our drug of focus, colistin, the AST revealed resistance of 41.1%, and other Egyptian studies have recorded extreme variations in colistin resistance–associated APEC. Awad et al. [18] reported a high incidence (92.31%) among 54 broiler flocks in two North Delta governorates. On the other hand, Moawad et al. [3] mentioned a low incidence (7.9%) from 48 broiler farms located in five governorates in northern Egypt. Additionally, the study addressed the isolated *E. coli* strains for colistin-associated resistance linked with the *mcr* genes. Special effort was made to detect the *mcr* genes responsible for the emergence of colistin resistance due to its importance as one of the last drugs of resort for the treatment of infections in humans.

All *E.coli* isolates (56/56, 100%) displayed dark pink to reddish colonies on the used chromogenic medium in the initial cultural screening, and this result is congruent with that mentioned by Abdul Momin et al. [28]. Nevertheless, only 14 isolates exhibited resistance to colistin (MIC ≥ 4 mcg/mL) with a percentage of 25% or higher by using the Colistin Ezy MIC strip (CL) (0.016- 256 µg/mL), as shown in Figure 2. In contrast to our data, a study of MIC testing that investigated 100 APEC strains was carried out in Iran and recorded 99% resistance against colistin [29].

Despite the two MIC test concentration differences, there was a close similarity in comparing the Sensititre test results with those of the E –test, and there was resemblance in 19 out of 20 tested isolates for colistin resistance (95%) (Table 3). These data were harmonized with those reported by Matuschek et al., in 2018 [30], and Kansak et al., in 2020 [31], who concluded that the use of commercial kits offered more accurate testing compared to the reference method and advised the use of such testing. The recent emergence of the concept of the genetic control of colistin resistance suggested the existence of both chromosomal mutations and transferable plasmid-mediated (*mcr*) genes [11]. All 56 *E. coli* isolates were tested for the presence of *mcr*1–5 gene variants by the application of conventional PCR assays and the data revealed that all isolates harbored only the *mcr*-1 gene and were negative for the other examined colistin resistance genes (*mcr*-2, *mcr*-3, *mcr*-4 and *mcr*-5), as demonstrated in Figure 3.

Our data agree with other reports which indicated that *mcr-*1 is the most important gene in colistin resistance [3,32]. Additionally, the same data have been reported in Northern Africa, Algeria [33], Morocco [34], Tunisia [35] and worldwide [36,37,38,39]. Interestingly, our data about the correlation between the existence of the *mcr*-1 gene and the other performed methods are greatly supported. The identical, 100% screening of chromogenic media and PCR was emphasized by Abdul Momin et al. [28], who mentioned that CHROMagar COL-APSE was more sensitive in supporting the growth of *Enterobacteriaceae* with colistin resistance when accompanied by the harboring of *mcr*-1.

Furthermore, the MIC results were in agreement with those reported by Liu, et al. [11], who mentioned that most of the *mcr*-1-harboring strains displayed a low MIC for colistin, at around 4 μg/mL. Furthermore, the presence of *mcr* genes does not always equate to colistin resistance phenotypically [40].

## 5. Conclusions

Pathogenic *E. coli* in poultry farms in Egypt constitutes a significant public health concern. This study demonstrated a high incidence of multidrug- and colistin-resistant *E. coli* isolates among the sampled poultry farms, with harboring of the *mcr*-1 gene (the major gene implemented in controlling colistin resistance, although the existence of *mcr* genes in a genome does not imply that the corresponding isolate is resistant). Phenotypic techniques are indicated as complementary indicators of the presence of colistin-resistant strains.

In our suggestion that the presence of the *mcr*-1 gene, combined with resistant MIC or the nearest zone of resistance, indicates that the bacteria harbor the resistance. The study findings revealed that, in countries where colistin is used for animal husbandry, a significant source of resistance in the human food chain could develop. The usage of commercial kits for colistin resistance–detection is more accurate, quicker and advised to be combined with the molecular techniques as a complementary diagnostic system. The findings also highlight the need to improve plans and programs for the proper use of antibiotics among poultry farms.

## Figures and Tables

**Figure 1 vetsci-09-00282-f001:**
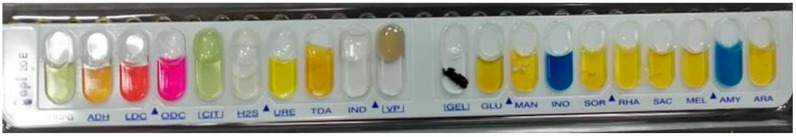
The biochemical profile of obtained *E. coli* isolates via API 20E system.

**Figure 2 vetsci-09-00282-f002:**
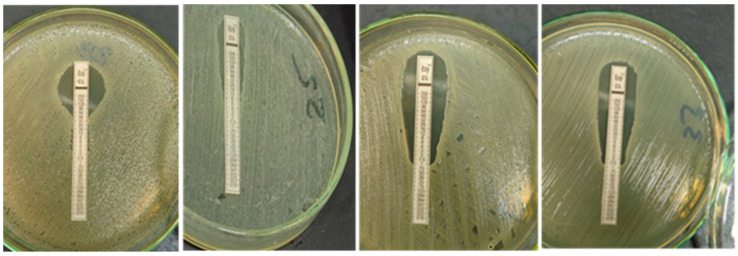
Colistin Ezy MIC strip (CL) showing different readings and configurations on Muller–Hinton agar plates.

**Figure 3 vetsci-09-00282-f003:**
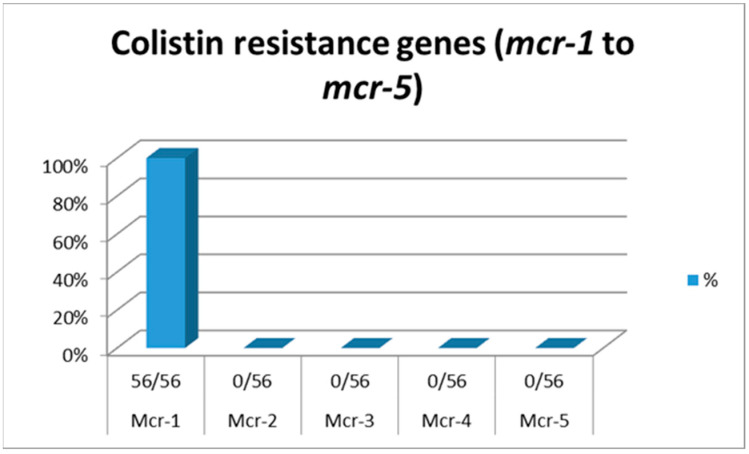
Data for colistin-resistance genes (*mcr-*1, *mcr-*2, *mcr-*3, *mcr-*4 and *mcr-*5).

**Table 1 vetsci-09-00282-t001:** Primer sequences, objective genes, amplicon sizes and cycling conditions.

Target Gene	Primers Sequences	Amplified Segment (bp)	Amplification (35 Cycles)			Final Extension	Reference
			Secondary Denaturation	Annealing	Extension		
***mcr-*1**	CGGTCAGTCCGTTTGTTC	309	94 °C 30 s	55 °C 30 s	72 °C 30 s	72 °C 10 min	[11]
	CTTGGTCGGTCTGTAGGG						
***mcr-*2**	TGGTACAGCCCCTTTATT	1617	94 °C 30 s	55 °C 40 s	72 °C 1.2 min	72 °C 12 min	[12]
	GCTTGAGATTGGGTTATGA						
***mcr-*3**	TTGGCACTGTATTTTGCATTT	542	94 °C 30 s	50 °C 40 s	72 °C 45 s	72 °C 10 min	[13]
	TTAACGAAATTGGCTGGAACA						
***mcr-*4**	ATTGGGATAGTCGCCTTTTT	487	94 °C 30 s	54 °C 40 s	72 °C 45 s	72 °C 10 min	[14]
	TTACAGCCAGAATCATTATCA						
***mcr-*5**	ATGCGGTTGTCTGCATTTATC	1644	94 °C 30 s	50 °C 40 s	72 °C 1.2 min	72 °C 12 min	[15]
	TCATTGTGGTTGTCCTTTTCTG						

**Table 2 vetsci-09-00282-t002:** Antimicrobial sensitivity patterns of isolated *E. coli* from poultry.

Antimicrobial Agent	*E. coli* (56) Poultry Isolates		
	Resistant No. (%) *	Intermediate No. (%) *	Sensitive No. (%) *
Ampicillin (AMP^10^)	52 (92.9%)	4 (7.1%)	0 (0%)
Apramycin (APR^30^)	40 (71.4%)	7 (12.5%)	9 (16.1%)
Ciprofloxacin (CIP^5^)	37 (66.1%)	7 (12.5%)	12 (21.4%)
Clindamycin (DA^2^)	56 (100%)	0 (0%)	0 (0%)
Colistin Sulphate (CT^10^)	23 (41.1%)	0 (0%)	33 (58.9%)
Norfloxacin (NOR^10^)	37 (66.1%)	4 (7.1%)	15 (26.8%)
Spectinomycin (SPT^100^)	31 (55.2%)	0 (0%)	25 (44.6%)
Streptomycin (S^10^)	54 (96.4%)	0 (0%)	2 (3.6%)
Sulfamethoxazole–Trimethoprim (SXT^25^)	46 (82.1%)	1 (1.8%)	9 (16.1%)
Tetracycline (T^30^)	56 (100%)	0 (0%)	0 (0%)

***** Percentage of positive samples.

**Table 3 vetsci-09-00282-t003:** Comparison of MIC values determined by Sensititre with those of E-Test.

Code No.	E-Test (µg/mL)	Interpretation	Sensititre (µg/mL)	Interpretation
1	1	S	≤0.25	S
2	4	R	=2	S
3	1	S	≤0.25	S
4	2	S	=2	S
5	1	S	≤0.25	S
6	4	R	=4	R
7	1.5	S	≤0.25	S
8	1	S	≤0.25	S
9	4	R	˃4	R
10	1	S	=0.5	S
11	1.5	S	≤0.25	S
12	1.5	S	≤0.25	S
13	1	S	≤0.25	S
14	1	S	≤0.25	S
15	4	R	=4	R
16	1	S	=0.5	S
17	2	S	=2	S
18	0.75	S	≤0.25	S
19	0.75	S	=1	S
20	4	R	˃4	R

(S) referred to sensitive while (R) referred to resistant to colistin.

## Data Availability

Data available in a publicly accessible repository.

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
