# Peer review of "Phenotypic and Genotypic Screening of Colistin Resistance Associated with Emerging Pathogenic Escherichia coli Isolated from Poultry"

_vetsci, 2022, doi:10.3390/vetsci9060282_

Round 1
Reviewer 1 Report
After the revisions and modifications performed by the authors, I consider that this manuscript could be published in its present form.
Author Response
please see the attached file for correction according to the reviewer’s comments.
I would like to inform you the above abstract was decreased in the number of its words to 200 as requested before

Reviewer 2 Report
Dear Authors
your manuscript is relevant for Public Health, due to the extensive use of colistin in humans and highlights the need to control of this antimicrobials in veterinary field. You have improved remarkably your work especially in discussion section . A last suggestion ....if you like. Something is wrong in title ...associated in emergence to . I think it is better "associated with emerging pathogenic E.coli "or "associated with emergence of pathogenic E.coli".
Author Response
Phenotypic and Genotypic screening of Colistin Resistance Associated with Emerging Pathogenic E. coli isolated from poultry.
we change the title as the reviewer requested.
thank you for your effort

This manuscript is a resubmission of an earlier submission. The following is a list of the peer review reports and author responses from that submission.
Round 1
Reviewer 1 Report
The reviewer's comments and suggestions for authors are in the attached file.

Reviewer 2 Report
Dear Authors
Although the colistin -resistance displayed by microrganisms, particularly both pathogenic and commensal E.coli, is really stimulating issue for its impact on Public Health, your manuscript shows a number of weakness points and thus not suitable for publication. In attachment my comments to the manuscript

Reviewer 3 Report
The paper by Badr H. et al., shows laboratory test results on 56 strains of poultry isolated E. coli and presents technical suggestion in order to diagnose colistin resentence accurately. Althought the Introduction is interesting, Abstract, Results, Discussion, and Conclusions sections are written in poor English grammar, and some of the sentences are difficult to understand.
Reviewer’s comments
Abstract
Line 42: Please clarify what ‘greatly confirmed’ mean.
Line: 147-148: Do the authors mean ‘High Pure PCR Template Preparation Kit’ from Roche instead of ‘Purification Kit’?
Line 147-152: The authors write that genomic DNA was prepared using the Kit above. However, the authors also cite in lines 78-79 that colistin-resistance genes are found not only in the chromosome but also on plasmids. Why were plasmid genes not investigated for colistin-resistance genes separately from the chromosomal genes?
Results:
Lines 204-207: The paragraph seems to be an explanation which should be included in the materials and methods section. The results for MIC by sensititre, as it says on the title for 3.2.4., is missing.
Figure 3, Lines 153-155: Why were mcr-5 to -9 genes excluded from the study?
Discussion:
Lines 228-239: The text is very difficult to follow because the paragraphs contain only one, or if not few, sentences. Why are the descriptions explaining serotypes divided into two paragraphs?
Lines 249-252: What is the meaning of the sentence: ‘Special valuation - - – last drugs of resort for humans infections’ therapy’? Please clarify.
Lines 260-264: The authors write that there was ‘close similarity’ with the results obtains by the sensititre test and the E-test. Since there is not results for the sensititre test presented, it is not possible to review this paragraph.
Lines 265-271: The paragraphs only contain one sentence each, and the text is very difficult to follow. Why are the descriptions explaining serotypes divided into two paragraphs?
Lines 259: Please clarify the reason for citing reference 30 in connection with the current results?
Lines 273, 282, Lines 287-289: The authors emphasize that they and others demonstrated that the mcr-1 gene was the most important gene in colistin resistance, but explains that ‘the presence of mcr genes does not always equate to colistin resistance’, and ‘existence of mcr genes in a genome does not imply that the corresponding isolate is resistant’. Please clarify the meaning. From the presented results, all 56 strains tested were mcr-1 positive (100%), and CHROMagar positive (100%). Do the authors mean that not all strains were colistin resistant?
Line 295: Please clarify what ‘more accurate’ mean. Throughout the manuscript, no positive controls or standards are demonstrated. Positivity in colistin-resistance was 41.1% by the disk test, and 25% by the E-test, compared to the 100% in CHROMOagar test and 100% in mcr-1 genomic DNA PCR. Do the authors mean that 100% of all 56 strains are colistin resistant, and that the other two tests have high false-negative rates?
Conclusion:
Lines 291-292: The authors are suggesting to combine mcr-1 gene with other tests for accurate measurements on colistin resistance, but objective data for the explanation is missing. Which of the tests gives the highest true-positive results and the least false-positive, false-negative results? Please demonstrate the effect of combining tests, in comparison of mcr-1 test alone, statistically.
Lines 293-294: Which commercial kits do the authors mean? Please clarify.
Minor comments:
Abstract
Line 146: E. coli should be in italic.
Lines 192, 257: Figure 2, Table 2 should be written with a capital in the begging, and without brackets.
References, Lines 397-408:
Reference 38: Genes should be written in italic.
Reference 39, 40: Capital letters are used only at the beginning of the sentence in the other references.
Reference 41: Reference 41 and reference 11 are the same.
Reference 42: The title is written twice.